# Melt Spinning Process Optimization of Polyethylene Terephthalate Fiber Structure and Properties from Tetron Cotton Knitted Fabric

**DOI:** 10.3390/polym15224364

**Published:** 2023-11-09

**Authors:** Nanjaporn Roungpaisan, Natee Srisawat, Nattadon Rungruangkitkrai, Nawarat Chartvivatpornchai, Jirachaya Boonyarit, Thorsak Kittikorn, Rungsima Chollakup

**Affiliations:** 1Department of Textile Engineering, Faculty of Engineering, Rajamangala University of Technology Thanyaburi, Pathum Thani 12110, Thailand; nanjaporn_r@rmutt.ac.th (N.R.); natee.s@en.rmutt.ac.th (N.S.); 2Department of Textile Science, Faculty of Agro-Industry, Kasetsart University, Bangkok 10900, Thailand; nattadon.r@ku.ac.th (N.R.); mill.nawarat@gmail.com (N.C.); 3Kasetsart Agricultural and Agro-Industrial Product Improvement Institute (KAPI), Kasetsart University, Bangkok 10900, Thailand; aapjab@ku.ac.th; 4Division of Physical Science, Faculty of Science, Prince of Songkla University, Songkhla 90110, Thailand; thorsak.k@psu.ac.th

**Keywords:** decolorization, melt spinning, phosphoric acid, recycle PET, TC fabric, winding speed

## Abstract

Polyester/cotton fabrics with different proportions of Tetron Cotton, TC (35% Cotton/65% PET), and Chief Value Cotton, CVC (60% Cotton/40% PET), were investigated by removing the cotton component under various phosphoric acidic conditions including the use of cellulase enzymes. The remaining polyethylene terephthalate (PET) component was spun using the melt spinning method. Only 85% H_3_PO_4_-Enz_TC could be spun into consistent filament fibers. The effects of Acid-Enz TC (obtained from a powder preparation of 85% H_3_PO_4_-Enz_TC) at different weight amounts (1, 2, 5, and 10 %wt) blending with WF-*r*PET powder prepared by white recycled polyester fabric were evaluated for fiber spinnability at different winding speeds of 1000 and 1500 m/min. The results revealed that recycled PET fiber spun by adding Acid-Enz_TC up to 10 %wt gave uniformly distributed filament fibers. A comparative study of the physical, thermal, and mechanical properties also investigated the relationship between the effect of Acid-Enz_TC and the structure of the obtained fibers. Acid-Enz_TC:WF-*r*PET (5:95) was the optimal ratio. The thermal values were analyzed by DSC and TGA and crystallinity was analyzed by XRD, with mechanical strength closed to 100% WF-*r*PET. The FTIR analysis of the functional groups showed the removal of cotton from the blended fabrics. Other factors such as the Acid-Enz_TC component in WF-*r*PET, extraction conditions, purity, thermal, chemical, and exposure experiences also affected the formability and properties of recycled PET made from non-single-component raw materials. This study advanced the understanding of recycling PET from TC fabrics by strategically removing cotton from polyester–cotton blends and then recycling using controlled conditions and processes via the melt spinning method.

## 1. Introduction

The global production of textile fibers has almost doubled over the past 20 years, from 58 million metric tons in 2010 to 109 million metric tons in 2020 [1]. The United States and the European Union discard 22 million tons of textiles every year, with only 15% recycled or donated and the rest ending up in landfills [2]. Only 1% of clothing materials is recycled, while 74% of post-consumer textiles are suitable for recycling. Nowadays, people throw away unwanted clothes instead of donating them, with less than half reused or recycled and only 1% recycled into new clothes. Technologies that enable clothes to be recycled into fibers are now starting to emerge. In 2021, two types of fibers—polyethylene terephthalate (PET) with a volume share of 54% and cotton with a volume share of 22%—dominated the composition of consumer textile fibers, together accounting for 88% of global textile fiber production [3] due to their relatively low outstanding costs performance and additional features for their intended use, such as breathability, stretchability, washability, or texture. PET, or polyester, is the most commonly used textile thermoplastic fiber. In 2019, polyester accounted for 52% of all fibers produced, while recycled polyester accounted for just 14% of the total. Most recycled fibers come from PET bottles and other commodities such as colorless and non-textile goods because polyester fiber decolorization remains one of the most important obstacles in textile-to-textile recycling [4].

Among clothing textiles, mixed fibers of PET and cotton such as Chief Value Cotton (CVC; with cotton making up over 50% of the blend) or Tetron Cotton (TC; with PET making up over 50% of the blend) predominate. Combining polyester fibers with cotton fibers gives low-cost cool clothing products that dry quickly, retain their shape, and are not easily wrinkled. Before recycling, these material combinations require careful sorting and mechanical separation [5]. However, these methods were not efficient in separating different types of fibers that were twisted together to form yarn during the spinning process. Chemical fiber separation techniques are applied using different chemo-physical behaviors to dissolve PET and cotton in different solvents [6] under diverse conditions. Selectively degrading cotton fibers can recover intact polyester fibers [7] for mechanical recycling as in fiberfill and nonwovens. To recycle PET from PET powder obtained after cotton separation, a gentle reagent, such as biodegradable enzymes (cellulase enzyme) [8] or phosphoric acid, was employed as a solvent to dissolve crystalline cellulose and facilitate subsequent regeneration during melt spinning [9]. Phosphoric acid is a non-corrosive and non-toxic chemical for the pretreatment of textiles [10] that can dissolve crystalline cellulose at atmospheric pressure under a moderate temperature. The research team designed a TC fiber treatment process by combining a process using phosphoric acid and enzymes [11] to dissolve cotton fibers from polyester fibers to ensure integrity.

In the quest to find sustainable solutions for recycling the substantial surplus of polyester fabric produced by the textile industry, this initiative departed from traditional recycling methods. Recycled PET is typically derived from bottles and can be transformed into either bottles or fiber grades through a series of modifications. The primary objective was to control critical *r*PET properties including viscosity, purity, ash content, and moisture levels.

Previous research [12] concentrated on developing techniques and optimal conditions using the “Fiber/Fabric to Fiber/Fabric” concept to produce recycled polyester fibers (*r*PET) from 100% white polyester fabric waste using a single-component recycling process. This method involved thermal and mechanical preparation through compression and grinding. Essential aspects were temperature and heat management, which included the precise regulation of time, temperature, and residence time while avoiding high-shear force processes. Ultimately, this innovative method proved effective for melt-spinning *r*PET fibers derived from PET fabric.

This study addressed the fabrication of recycled polyester fibers from blended yarn of cotton–polyester (TC or CVC fabrics, non-single component) as received from industrial waste polyester raw materials, compared to 100% polyester (single component) by a cotton removal method under acidic and enzymatic conditions. Fiber/Fabric to Fiber/Fabric was the main concept of this research. Compared to the traditional method (Bottle to Fiber/Fabric), more complicated mechanisms of modifying Fiber/Fabric-grade polyester for use in the fiber-forming process involved the pretreatment of the blended yarn of cotton–polyester (TC fabrics) using different proportions of acid and enzymatic pretreatment to extract only PET fibers for the recycling process [8,10]. Grinding PET fibers to PET powders and drying before melt spinning [12] were also studied. The *r*PET fibers were spun using different proportions of extracted PET powders from white TC fabrics, with the blending of pure PET powder obtained from white recycled polyester fabric (as described [12]). The spinnability of these materials during the fiber formation process was conducted at different winding speeds. The fiber characteristics of the *r*PET blended fibers including their morphology, physical properties, thermal properties, mechanical properties, and functional groups of the *r*PET blended fiber surfaces were compared to bottle PET fibers.

## 2. Materials and Methods

### 2.1. Materials

White TC knitted fabric of 35% cotton/65% PET (code WF_TC) at basis weight 250 g/m^2^ and white CVC knitted fabric of 60% cotton/40% PET (code WF_CVC) at basis weight 180 g/m^2^ sponsored from Yong Udom Karn Tho Co., Ltd. (Bangkok, Thailand) and L.V.W. Group Co., Ltd. (Nakhon Pathom, Thailand), respectively, were used as textile wastes to recover PET. White PET knitted fabric (code WF_PET) at basis weight 136.87 g/m^2^ obtained from the stock of Jong Stit Co., Ltd. (Bangkok, Thailand), while recycled bottle PET (code BO_PET) sponsored by Teijin Co., Ltd. (Bangkok, Thailand) were used as control samples. All fabrics were defibrillated using a recycling machine.

### 2.2. Cotton Removal with Phosphoric Acid Pretreatment on Different Textile Waste Blends

Different defibrillated textile wastes blended with cotton and PET 35/65 (code TC) and 60/40 (code CVC) were studied for cotton removal using phosphoric acid pretreatment. Defibrillated TC fabric was pretreated using 85% phosphoric acid with codes of H_3_PO_4__TC or H_3_PO_4__CVC at LR 1:15 (*w*/*v*) at 50 °C for 7 h following the method of [11] to hydrolyze cotton from TC fabric into solution. The hydrolysis step was stopped by rapid dilution with 400 mL of deionized water. The crude PET was separated using a 20 mesh sieve and washed using deionized water to remove the cellulose. The remaining PET was oven-dried for 12 h at 60 °C.

Fourier transform infrared (FTIR) analysis of the remaining PET was applied to characterize the functional groups of polymers within the cotton/PET textile waste blend before and after pretreatment. An ATR-FTIR spectrometer (Nicolet IR200 FTIR, Thermo Scientific, Madison, WI, USA) was used over the range 400–4000 cm^−1^ with spectral resolution of 4 cm^−1^ and 128 scans.

Thermogravimetric analysis (TGA) was applied to assess the thermal properties of the fiber in each step from 25 to 500 °C. All remaining PET samples from the two textile wastes and bottle PET grade were measured using a thermogravimetric analyzer (Mettler Toledo, TGA/SDTA 851e, Zurich, Switzerland). The samples were weighed (2–5 mg) and heated at a rate of 10 °C/min under a nitrogen atmosphere.

Differential scanning calorimetry (DSC) was achieved for running the remaining PET from the two textile wastes and bottle PET grade. The melting, crystallization, and glass transition behavior of *r*-PET filaments from the different *r*-PET powders were studied using a differential scanning calorimeter (DSC, Model 200F3, Netzsch, Burlington, VT, USA). Each specimen was heated in an aluminum pan from room temperature to 200 °C at a rate of 10 °C/min and then cooled to 0 °C. 

### 2.3. Cotton Removal from TC with Phosphoric Acid and Enzymatic Pretreatment on TC Fabrics

Defibrillated TC fabric was pretreated at varied concentrations of phosphoric acid (55–85% by *v*/*v*) with codes of H_3_PO_4_-Enz_TC at LR 1:15 (*w*/*v*) at 50 °C for 7 h following a previous method [11] to hydrolyze cotton from TC fabric into solution. The hydrolysis step was stopped by rapid dilution with 400 mL of deionized water. The crude PET was separated using a 20 mesh sieve and washed using deionized water to remove the cellulose. The crude PET was then pretreated with cellulase at a dosage of 25 FPU/g substrate in citric buffer (pH 4.8) at 50 °C and 200 rpm for 96 h to remove the remaining cellulose. The reaction was stopped by filtration and water washing, with the remaining PET fibers oven-dried for 12 h at 60 °C.

An ATR-FTIR spectrometer (Nicolet IR200 FTIR, Thermo Scientific, Madison, WI, USA) was used to characterize the functional groups of polymers within the cotton/PET textile waste blends before and after pretreatment over the range 400–4000 cm^−1^ with a spectral resolution of 4 cm^−1^ and 128 scans.

The PET fiber morphology after hydrolysis treatment for each sample was investigated under a light microscope (Leica, LM750, Singapore).

### 2.4. Melt Spinning and Characterization of rPET Fibers Prepared from the Remaining PET Fibers in TC Fabric

The remaining PET fibers from Section 2.3 with 85% phosphoric acid and cellulase treatment (coded Acid-Enz_TC) were softened by heat compression at 230 °C for 3 min and then at 250 °C for 5 min before cooling to room temperature for 5 min to solidify the molten compressed sample. The PET samples were then ground to powder for melt spinning. A rheometer (C.B.N. Engineering Ltd., Bangkok, Thailand) was used to measure the melt flow index following ASTM D 1238 [12].

Before melt spinning, the Acid-Enz_TC was preheated and dehumidified under a hot air oven at 140 °C for 1 h. Preheating and completion of the crystalline phase were necessary to prevent solid-state formation during heating. The *r*PET powder was prepared using a melt spinning machine (Thermo Haake Polydrive, Karlsruhe, Germany). The *r*PET from TC fabric treatment (code Acid-Enz_TC) at 1–10% was blended with PET fabric sample (code WF_*r*PET) from a previous study [13] before melt spinning to the filament. The spinneret was set at 255–260 °C with a throughput rate of 0.24 g/hole/min, with draw ratios of 1000 and 1500 m/min (Table 1).

The *r*PET fibers prepared from *r*PET powder of both remaining Acid-Enz_TC/WF_*r*PET blended compounds at 1/99, 2/98, 5/95, and 10/90 and 0/100 samples at 1000 m/min winding speeds were characterized as follows.

#### 2.4.1. Differential Scanning Calorimetry (DSC)

The melting, crystallization, and glass transition behavior of different ratios of Acid-Enz_TC/WF_*r*PET blended compounds were studied using a differential scanning calorimeter (DSC, Model 200F3, Netzsch, Burlington, VT, USA). The specimen was heated in an aluminum pan from room temperature to 200 °C at a rate of 10 °C/min and then cooled to 0 °C. Heat history was recorded before reheating to 200 °C at the same rate. Cold crystallization and melting temperatures (T_c_ and T_m_, respectively) were obtained from the DSC curves. Delta *H_m_*, as the melting enthalpy (J/g), was used to calculate the percentage crystallinity, as shown in Equation (1):(1)$$\%Xc=\frac{∆H_{m}}{∆{H_{m}}^{0}}\times 100\%$$
where Δ*H_m_* [J/g] is the peak area (melting enthalpy) and Δ*H_m_*^0^ [J/g] is the melting enthalpy of a perfect PET crystal, equal to 140.1 J/g [14].

#### 2.4.2. X-ray Diffraction Analysis (XRD)

XRD patterns of different ratios of Acid-Enz_TC/WF_*r*PET blended compounds were achieved using a Bruker D8 Advance model with CuKα radiation at a wavelength of 1.54 Å. The diffraction spectrogram was recorded as 2θ (2 theta) ranging from 5 to 80 °C.

#### 2.4.3. Thermogravimetric Analysis (TGA)

TGA was used to determine the thermal properties of different ratios of Acid-Enz_TC/WF_*r*PET blended compounds in each step from 25 to 800 °C. All *r*PET filaments were measured by a thermogravimetric analyzer (Mettler Toledo, TGA/SDTA 851e, Switzerland). The samples were weighed (2–5 mg) and heated at a rate of 10 °C/min under nitrogen gas at a constant flow rate of 20 mL/min.

#### 2.4.4. Fiber Morphology under an Optical Microscope

Cross section and longitudinal section fiber morphologies of different ratios of Acid-Enz_TC/WF_*r*PET blended compounds were studied after melt spinning under an optical microscope (Olympus Model GX41, Tokyo, Japan). Fiber diameter was measured using software and fiber fineness (denier) was calculated using Equation (2):(2)$$\mathrm{Mass}=\mathrm{Denier}=\rho \;\times \;\frac{\pi a^{2}}{4}\;\times \;900,000$$
where denier is a direct measure of linear density as substance mass per unit volume (g/cm^3^) and a is the diameter (cm). In Equation (3):(3)$$\rho =\frac{\mathrm{mass}}{\pi (\frac{a^{2}}{2})\times L}$$
where mass (g) is defined as the amount of matter in a substance, a is the diameter (cm), and L is the length (cm).

#### 2.4.5. Mechanical Properties

Tensile strength and percentage elongation at break of different ratios of Acid-Enz_TC/WF_*r*PET blended compounds were measured following the standard method of ASTM D3822 [15] using a tensile tester (Instron Model 5560, Grove City, PA, USA).

## 3. Results

### 3.1. Cotton Removal with Phosphoric Acid Pretreatment on Different Textile Waste Blends

All PET textile wastes from different ratios of cotton and PET at 60:40 (code CVC) and 35:65 (code TC) were defibrillated using a recycling machine, before pretreating with phosphoric acid at 85% to dissolve the cotton part. The remaining PET in all samples was then analyzed by FTIR spectroscopy. For the CVC fabric, the structure of the cotton was shown mainly at 3342 cm^−1^ attributed to the O-H stretching vibration of cellulose [16], and at 2917 cm^−1^ attributed to the C–H stretching vibration of cellulose [17]. The FTIR peak at 1057 cm^−1^ was due to the –C–O–C pyranose ring skeletal vibration and at 897 cm^−1^ due to β-glycosidic linkages of cellulose [18]. The PET structure contributed mainly to the TC fabric at 1725 cm^−1^, attributed to carbonyl (C=O) stretching, 1265 cm^−1^, attributed to the C(=O)-O stretching of the terephthalate group, and 1102 cm^−1^, attributed to the C-O stretching of ethylene glycol [19]. Compared to the remaining PET component after phosphoric acid pretreatment on the two textile wastes, a distinctive PET peak of H_3_PO_4__TC was found in the range 1700–1102 cm^−1^ with no peaks of cellulose remaining. The results showed that conditions for dissolving cotton apart from PET should be carried out with a cotton content of less than 50%, which is suitable for dissolving TC to reuse as *r*PET. For reusing CVC, conditions for cotton dissolving involved using a phosphoric acid treatment 2–3 times (Figure 1).

The thermal properties of the remaining PET fibers in treated CVC or TC fabrics after phosphoric acid pretreatment (85% H_3_PO_4__CVC or 85% H_3_PO_4__TC) are shown in Figure 2. The thermal stability of 85%H_3_PO_4__TC was similar to WF_PET (knitted PET fabric as the reference). Phosphoric acid pretreatment at 85% completely hydrolyzed the cotton component in TC.

Meanwhile, 85% H3PO4_CVC with a 60% cotton content retained some cotton components, which decomposed between 280 and 400 °C owing to the acetyl group in hemicellulose [20] and between 310 and 400 °C due to cellulose decomposition [20,21]. Hence, the substantial presence of cotton fibers hindered full dissolution in H_3_PO_4_, leading to an investigation of TGA properties, revealing degradation caused by cellulose within the remaining structure. Thus, the WF_TC was selected to study the effect of acid and enzyme pretreatment for the complete hydrolysis of cotton.

The thermal properties were analyzed by DSC, with both the absorption and exothermic thermograms presented in Figure 3. The thermal behavior of WF_PET was compared to H_3_PO_4__TC (phosphoric-acid-treated TC). The remaining PET fibers (H_3_PO_4__TC) had Tm and Tc values comparable to PET fibers obtained from white knitted fabrics of PET (WF_PET). The remaining PET fibers from H_3_PO_4__TC were used to prepare PET powder by a compression machine at the melting temperature of PET and then cooled to room temperature to prepare the PET sheet before grinding into powder for melt spinning. After compression, the H_3_PO_4__TC sheet was burned at the edge because some cotton remained in the components. The phosphoric acid method did not completely remove the cotton in TC. Therefore, an enzymatic treatment was applied after phosphoric acid pretreatment following previous research [8].

### 3.2. Cotton Removal from TC with Phosphoric Acid and Enzymatic Pretreatment on TC Fabrics

Figure 4 presents the percentage hydrolysis yield of different concentrations of phosphoric acid and the cellulase enzyme and the fiber’s appearance under an optical microscope. A concentration of 85% phosphoric acid and enzyme pretreatment were used to remove cellulose from the TC fabric, as shown by the percentage hydrolysis yield and fiber appearance under the microscope. The ATR-FTIR spectrum (Figure 5) confirmed that no peaks were attributed to the cellulose functional groups at wavenumber ranges of 3500 to 3200 cm^−1^, 2923 cm^−1^, and 2854 cm^−1^. A pronounced peak of PET appeared at a wavenumber in the range of 1700–1102 cm^−1^. The 85% H_3_PO_4_-Enz_TC was hot pressed with compression at 250 °C without burning, and then ground to a powdered state using a combination of compression molding and mechanical grinding. This process was also applied to white PET fabric samples (WF_PET), as studied by [13]. Subsequently, both Acid-Enz_TC and WF_PET were spun by the melt-spinning process with different ratios (1, 2, 5, and 10% of Acid-Enz_TC) to produce an *r*PET uniform filament.

### 3.3. Melt Spinning and Characterization of rPET Fibers Prepared from Remaining PET Fibers in Treated TC Fabric

For the melt-spinning process of two types of *r*PET powders derived from white fabric and TC fabric waste, the predominant component was WF_*r*PET powder, which was blended with varying percentages of Acid_Enz_TC powder. The maximum allowable loading capacity and winding speed were set at 10% Acid_Enz_TC + 90% WF_*r*PET and 1500 m/min, respectively, due to limitations in spinnability. In processing, it is imperative to meticulously adjust key parameters, including the throughput rate, melting temperature, and winding speeds. This adjustment is critical to achieve consistent fiber formation with optimal performance (Table 1).

Within this experiment, the spinning temperature deviated from the standard range of standard polyester grades by 10 to 15 °C. The temperature at the spinning head component was carefully maintained within 255–260 °C to prevent some degradation effects. Deviating from this prescribed temperature range resulted in excessive fluidity of the molten fibers and unfeasible conditions, as shown in Figure 6. When mixing a quantity of Acid_Enz_TC exceeding 20% by weight, the molten polymer rheology became irregular. The fibers were unable to maintain their shape and broke when subjected to high-speed stretching. Conversely, at concentrations of 1%, 2%, and 5% Acid_Enz_TC, continuous formation was achievable and fibers with consistent characteristics were obtained.

The melt flow index values for both categories of blended *r*PET powder are shown in Figure 6, providing essential guidance for establishing precise temperature settings during the fiber formation process. Various blending ratios between WF_*r*PET and Acid-Enz_TC were investigated for molten flow behavior related to viscosity, conducted at a controlled testing temperature of 260 °C. Elevating the proportion of Acid-Enz_TC from 1 to 10 %wt consistently led to a reduction in the flow rate and a concurrent increase in viscosity when compared to pure WF_*r*PET. This intriguing outcome was ascribed to the cellulose elimination process applied to TC fabric, which selectively retained only the polyester component. Consequently, this selectivity impacted the flow rate in distinctive ways, particularly as the blending ratio attained the threshold of 20%. When the quantity of Acid-Enz_TC was greater than 20%, the formation process became unfeasible and the resulting fibers exhibited a droplet-like morphology and experienced interruptions during the stretching process (Figure 6). This phenomenon was attributed to thermal degradation resulting from the extraction system. The melt spinning results of these blended ratios are shown in Table 2.

To conduct a comprehensive analysis of fiber uniformity, size, and fineness achieved from the production process, both pure WF_*r*PET and the blended samples were subjected to an optical microscope examination. The results demonstrated a consistent and uniform size distribution throughout the entire boundary (Figure 7). This consistency in size and evenness emphasized the success achieved in the fiber production process under the specified conditions. However, when comparing the outcome resulting from increasing Acid-Enz_TC contents, this augmentation significantly contributed to the noteworthy increase in fiber dimensions (Table 3). This specific phenomenon was attributed to a shift in melt flow values including the crystallization rate, as well as a reduced residence time due to thermal degradation [22], among a multitude of other influential factors.

Polyesters typically exhibit crystallization at specific peak values in X-ray diffraction (XRD) patterns at 16, 17.5, 22.5, and 25.5 degrees [23]. The XRD results, illustrated in Figure 8, showed that no peaks appeared. This observation was noteworthy when comparing the outcomes of various quantities of Acid-Enz_TC in the blend, conducted at a winding speed of 1000 m/min. Under these specific conditions, where the fiber cannot crystallize, the XRD analysis did not reveal any discernible peaks at any positions. The material appeared to remain in an amorphous state. Polyesters were characterized by a slow crystallization rate [24,25,26] and achieving crystallization may necessitate an annealing process with an elevated temperature and speed to enhance the crystallization rate.

The thermal stability properties of WF-*r*PET fiber samples that were either non-blended or blended with Acid-Enz_TC in varying quantities were determined by thermogravimetric analysis (TGA). The results showed no significant differences in the ability to withstand temperature increases within the range of 30–392 °C, as depicted in Figure 9. When comparing the stability, all samples maintained a higher remaining weight at temperatures exceeding 392 °C. WF-*r*PET 100% exhibited the highest tendency to retain weight compared to Acid-Enz_TC+WF-*r*PET.

Several factors contributed to this phenomenon. For instance, the white polyester fabric may contain trace additives that result in ash residuals, while blended fibers may exhibit fluctuations in weight loss due to the post-effects of the etching cellulose system that can alter some components and change physical properties. Moreover, the composition of the original fabric between 100% white polyester and TC fabric may differ in terms of properties and additive ratios, leading to variations in weight loss tendencies [27].

Figure 10 shows the heating and cooling thermograms of *r*PET fibers prepared from recycled PET materials with different sources, specifically *r*-BO_PET grade (commercial reference), white fabric (WF-*r*PET), and varying %Acid_Enz_TC in WF-*r*PET. These fibers were manufactured through melting in situ with the increase of the winding speeds for each ratios.

Figure 10a shows that all samples exhibited a cold crystallization temperature (Tcc) with some exothermic areas at approximately 124 °C, while a shift in this behavior occurred when the TC content exceeded 5%, as indicated by the appearance of a relatively substantial exothermic region. This observation agreed with the XRD data, suggesting that the fibers initially existed in an amorphous state. Subsequently, when exposed to temperatures higher than the glass transition temperature (Tg) during heating in DSC measurements, molecular alignment, as well as crystallization, occurred followed by a melting temperature of 264 °C, with only minor variations in the melting temperature among the samples. To provide further insights, *r*PET exhibited a relatively low crystallization rate. Consequently, incomplete structures or the amorphous phase led to recrystallization when exposed to temperatures higher than Tg, as indicated by the significant exothermic cold crystallization area [28].

Figure 10b presents the cooling thermograms as the temperature descended from the molten state. A notable crystallization event was observed in the fiber samples within the temperature range of 215–221 °C. For samples blended with 1–2% Acid_Enz_TC in WF_*r*PET, the position of TC did not exhibit significant deviations from the references (*r*-BO and WF-*r*PET 100%). Conversely, 5% Acid_Enz_TC in WF_*r*PET induced the crystallization rate or faster nucleation, accompanied by an increase in the enthalpy of crystallization. The graph also displayed a sharp characteristic signifying the presence of Acid_Enz_TC in WF_*r*PET crystallinity. This potentially impacted the fiber formation behavior spin-line or processability. Furthermore, thermal experiences encountered during the fiber formation process may have consequential effects on the resulting fiber properties.

The graphs obtained from the tenacity and percentage elongation testing of *r*PET fibers between pure WF-*r*PET and WF-*r*PET blended with Acid-Enz_TC at different quantities are shown in Figure 11. Significant factors supported the development of the fiber structure in terms of alignment along the axis and the occurrence of crystallinity [25]. An increase in the tenacity value occurred when the winding speed increased, while the percentage elongation decreased. Although the blending of Acid-Enz_TC in WF-*r*PET increased the crystallinity, this led to a reduction in the tenacity value due to the increased brittleness of the fibers. In industrial production, additional modifiers may need to be introduced to balance the flow properties and other factors to efficiently enable the production of recycled PET fibers.

As depicted in Figure 12, the FTIR spectra of WF-*r*PET fibers, both non-blended and blended with varying quantities of Acid-Enz_TC, at a winding speed of 1000 m/min confirmed that these fibers were *r*PET fibers, consistent with research conducted by [28], and exhibited similar graph characteristics. However, WF-*r*PET fibers blended with Acid-Enz_TC may still contain residual cellulose, as indicated by the FTIR results.

After effectively eliminating cellulose from the TC fabric through the application of an acid and enzyme system, the resulting Acid-Enz_TC fabrics were subjected to FTIR analysis to conclusively verify the presence of the PET component, as shown in Figure 12. The purified *r*PET materials extracted from the TC fabric were then converted into a powdered state using a combination of compression molding and mechanical grinding. This process was the same as that applied to white PET fabric samples (WF_*r*PET). Subsequently, both types of *r*PET, sourced from distinct origins, were subjected to the melt-spinning process with different ratios to produce uniform fibers.

## 4. Conclusions

In summary, this study focused on the conversion of a blend of cotton and polyester fabrics, constituting a “multi-component recycling process,” into usable *r*PET fibers. Previous researches had explored cellulose extraction from fabrics, but no subsequent steps were taken to reconvert this cellulose into fibers. In our experiment, TC (35% cotton/65% polyester) was employed to remove cotton components under varying phosphoric acid conditions, enhancing the *r*PET purity and yield. Enzyme pretreatment was also applied to degrade the remaining cotton after acid extraction. Significantly, only the combination of 85% H_3_PO_4_ and enzymes, referred to as 85%H_3_PO_4_-Enz_TC, enabled the continuation of fiber formation. The optimal condition for *r*PET fiber production involved melt spinning using 85% H_3_PO_4_-Enz_TC blended with pure *r*PET powder obtained from white recycled polyester fabric (WF_*r*PET) at a 5:95 ratio. The *r*PET displayed good stability, as indicated by its thermal profile via TGA and DSC, along with mechanical strength nearly comparable to 100% WF_*r*PET. These extensive analyses have provided valuable insights for further research into the recycling of multi-component polymer materials, particularly polyester–cotton blends, as surplus products in the textile industry.

## Figures and Tables

**Figure 1 polymers-15-04364-f001:**
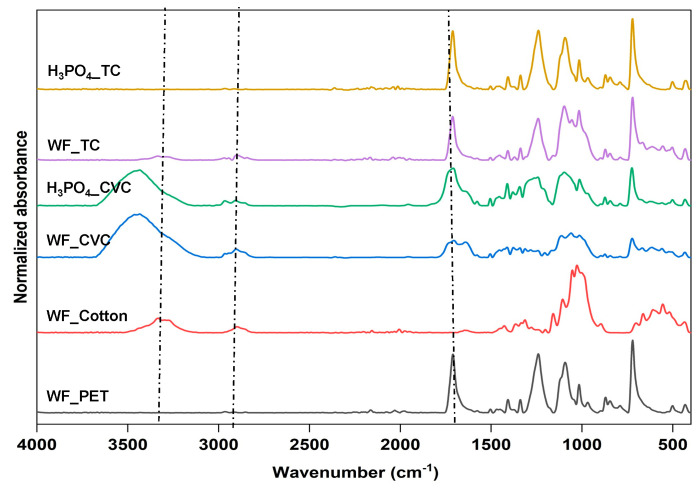
FTIR spectra of remaining PET fibers in treated CVC or TC fabrics before and after phosphoric acid pretreatment with WF_cotton and WF_PET as references. Dotted lines at 3342 and 2917 cm^−1^ attributed to the main function groups of cellulose; dotted line at 1725 cm^−1^ attributed to the main function group of PET.

**Figure 2 polymers-15-04364-f002:**
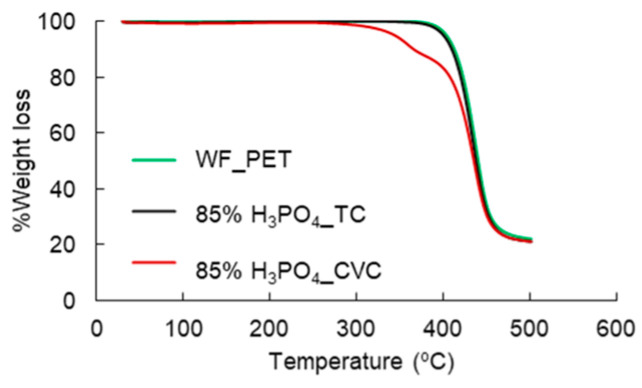
TGA spectra of remaining PET fibers in treated CVC or TC fabrics by phosphoric acid pretreatment with WF_PET as reference.

**Figure 3 polymers-15-04364-f003:**
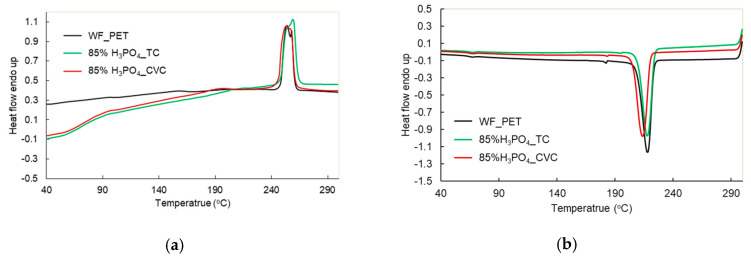
DSC thermograms of PET powder prepared from 85% H_3_PO_4__TC and 85% H_3_PO_4__CVC after recrystallization at different temperatures, (**a**) heating thermograms and (**b**) cooling thermograms.

**Figure 4 polymers-15-04364-f004:**
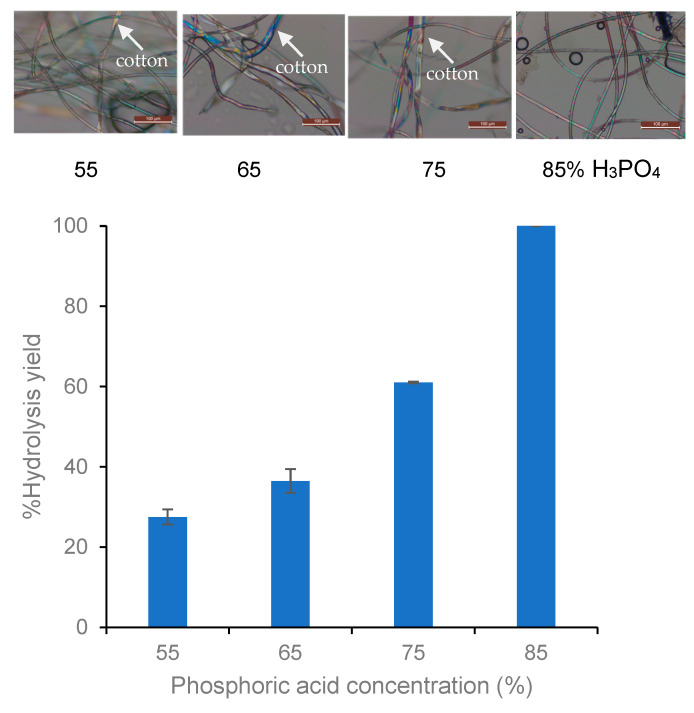
Percentage of hydrolysis yield and appearance of remaining fibers after phosphoric acid pretreatment under an optical microscope.

**Figure 5 polymers-15-04364-f005:**
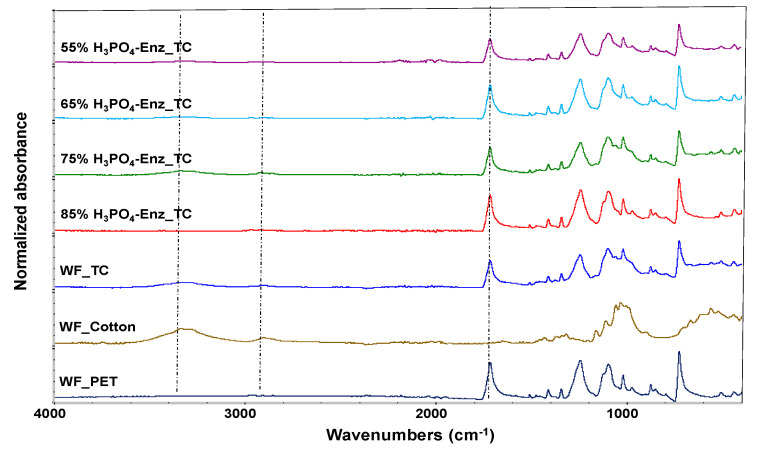
FTIR spectra of remaining PET fibers in TC fabrics using phosphoric acid and enzymatic pretreatment with WF_cotton and WF_PET as references. Dotted lines at 3342 and 2917 cm^−1^ attributed to the main function groups of cellulose; dotted line at 1725 cm^−1^ attributed to the main function group of PET.

**Figure 6 polymers-15-04364-f006:**
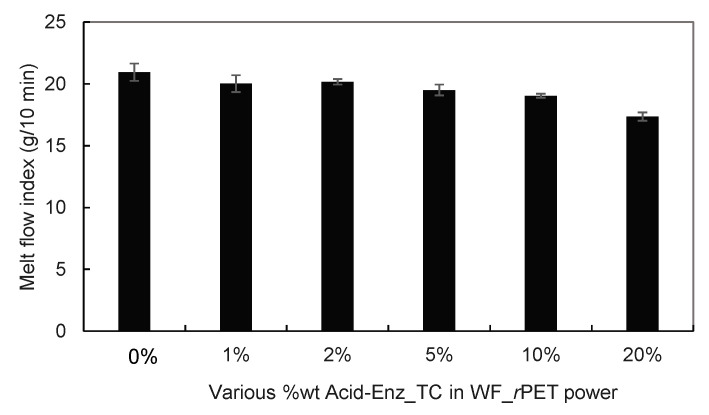
Melt flow index for *r*PET compared with varying amounts of Acid-Enz_TC in *r*PET, at a testing temperature of 260 °C.

**Figure 7 polymers-15-04364-f007:**
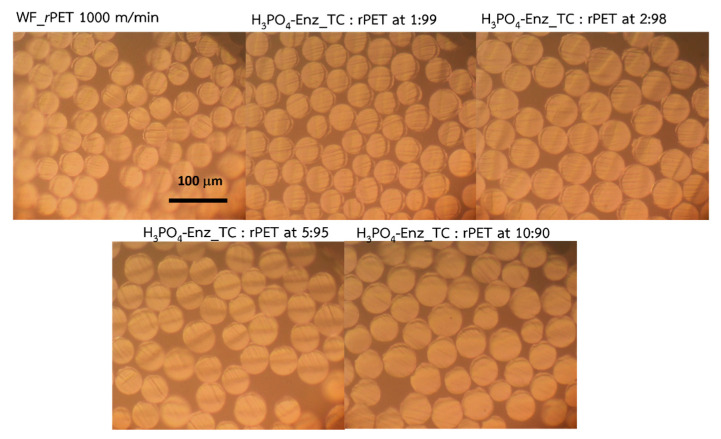
Cross-sections using optical microscopy of WF_*r*-PET fibers and blends with varying Acid-Enz_TC percentages of 1, 2, 5, and 10%, collected at a winding speed of 1000 m/min.

**Figure 8 polymers-15-04364-f008:**
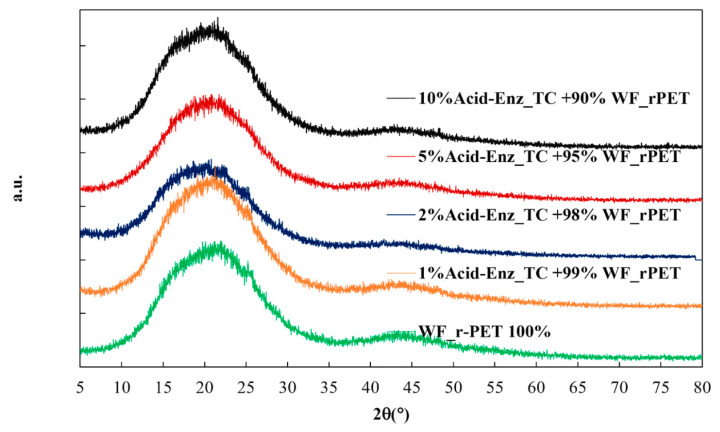
Crystalline structures using X-ray diffraction (XRD) analysis of WF_*r*PET fibers and blends with varying Acid-Enz_TC percentages of 1, 2, 5, and 10%, collected at winding speed of 1000 m/min.

**Figure 9 polymers-15-04364-f009:**
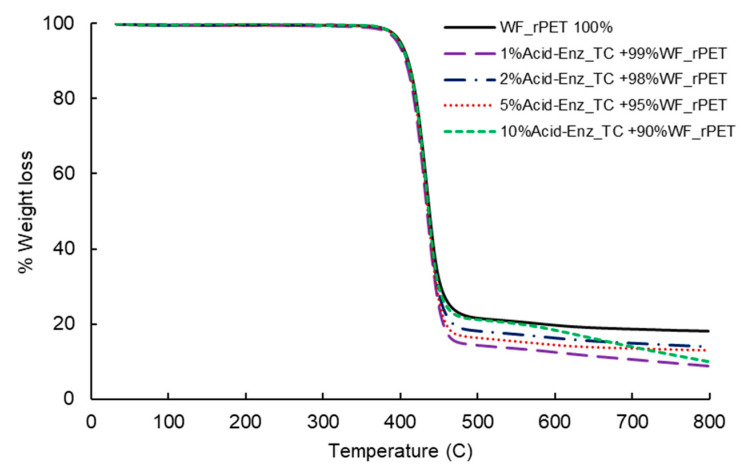
TGA thermogram of *r*PET fibers prepared from blends of Acid-Enz_TC and WF-*r*PET at different ratios at a winding speed of 1000 m/min.

**Figure 10 polymers-15-04364-f010:**
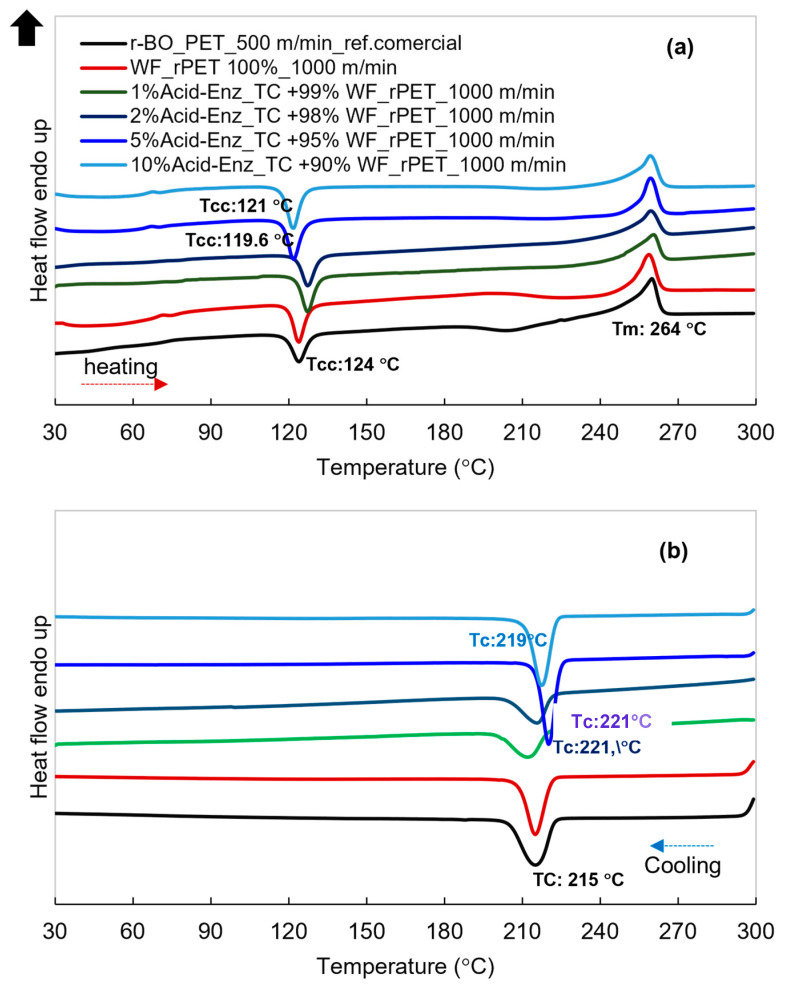
DSC results: (**a**) heating and (**b**) cooling thermograms of *r*PET fibers prepared from blends of H_3_PO_4_-Enz_TC and WF-*r*PET at different ratios and at winding speed of 1000 m/min (*r*_BO_PET is bottle-grade *r*PET as a commercial reference).

**Figure 11 polymers-15-04364-f011:**
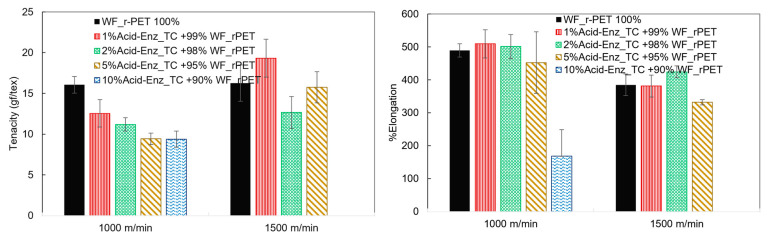
Tenacity and %elongation of *r*PET fibers prepared from blends of H_3_PO_4_-Enz_TC and WF-*r*PET at different ratios and at winding speeds of 1000 and 1500 m/min.

**Figure 12 polymers-15-04364-f012:**
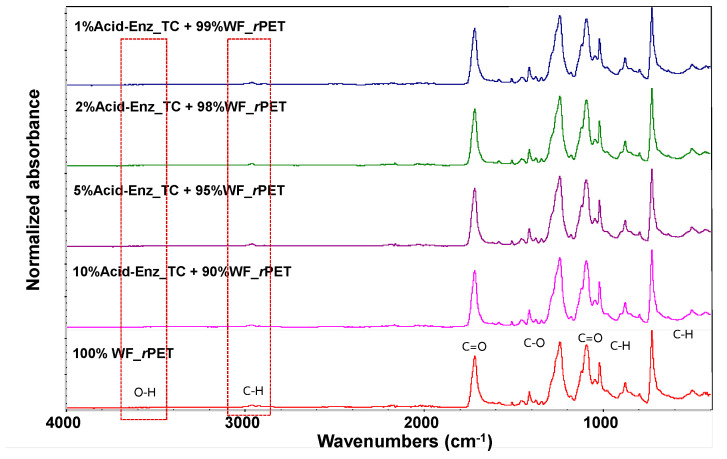
FTIR spectra of *r*PET fibers prepared from blends of Acid-Enz_TC WF-*r*PET at different ratios.

**Table 1 polymers-15-04364-t001:** Melt spinning parameters of *r*-PET powder prepared from Acid-Enz_TC and WF_*r*PET blended compounds at different ratios.

Parameter	Setting
Orifice configuration	Round (0.32 mm)
Spinning temperature (°C)	245/255/260/255–260
Through rate (g/hole/min)	0.24
Take-up speed (m/min)	1000 and 1500

**Table 2 polymers-15-04364-t002:** Processability of Acid-Enz_TC+WF_*r*PET with different ratios and winding speeds.

Sample Name	Winding Speed (m/min)	
	1000	1500
*r*PET_1000	/	/
1% Acid-Enz_TC + 99% *r*PET	/	/
2% Acid-Enz_TC + 98% *r*PET	/	/
5% Acid-Enz_TC + 95% *r*PET	/	/
10% Acid-Enz_TC + 90% *r*PET	/	Δ
20% Acid-Enz_TC + 80% *r*PET	X	X

**Table 3 polymers-15-04364-t003:** Fineness of *r*PET fibers prepared from different ratios of Acid-Enz_TC and WF_*r*PET.

Sample Name	Diameter (Micron)
100% WF_*r*PET	16.1 ± 0.92
1% Acid-Enz_TC + 99% WF_*r*PET	18.8 ± 0.99
2% Acid-Enz_TC + 98% WF_*r*PET	20.8 ± 0.74
5% Acid-Enz_TC + 95% WF_*r*PET	21.2 ± 0.79
10% Acid-Enz_TC + 90% WF_*r*PET	21.5 ± 1.74

## Data Availability

Data are contained within the article.

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
