# Peer review of "Melt Spinning Process Optimization of Polyethylene Terephthalate Fiber Structure and Properties from Tetron Cotton Knitted Fabric"

_polymers, 2023, doi:10.3390/polym15224364_

Round 1

Reviewer 1 Report

Comments and Suggestions for Authors

The paper is very interesting and contributes to sustainability in the meaning of recycling. Ther research is well structured, but there are lots of typos ie condi-tions in p.2,l.87 ; L.124, p.3 °C should be used instead 0 superscript, etc. ; odd spacing ie first paragraph in Introductions, full p.9, figure caption etc.

In some cases English is very poor so it is not understandable ….“So, enzymatic treatment needed to apply after phosphoric acid pretreatment following [8].“

Some issues should be addressed:

In Introduction, so many details of this study is not necessary (like %, speed, etc, just state at diferent conc and speed…put details in results and discussion).P.3 line 100 , “process e.g. Also th”…. Something is missing in these sentences! From that line further text should be rewritten!

All standards used should be listed in references, ie ASTM.

Conclusion should conclude findings, not elaborate the study. It is written more like an abstract than conclusion. Please, shorten it I make conclusive remarks only!

Comments on the Quality of English Language

English should be improved. In some cases it is not understandable. Please use English proofreading to improve

Author Response

Comments and Suggestions for Authors

The paper is very interesting and contributes to sustainability in the meaning of recycling. Ther research is well structured, but there are lots of typos ie conditions in p.2,l.87 ; L.124, p.3 °C should be used instead 0 superscript, etc. ; odd spacing ie first paragraph in Introductions, full p.9, figure caption etc.

Response: The authors thank you for pointing this out. We agree with this comment. Therefore, we corrected all typos as following of reviewer’s comments.

In some cases English is very poor so it is not understandable ….“So, enzymatic treatment needed to apply after phosphoric acid pretreatment following [8].“

Response: The authors thank you and corrected English all manuscript as shown in yellow highlight. 

Some issues should be addressed:

In Introduction, so many details of this study is not necessary (like %, speed, etc, just state at diferent conc and speed…put details in results and discussion).P.3 line 100 , “process e.g. Also th”…. Something is missing in these sentences! From that line further text should be rewritten!

All standards used should be listed in references, ie ASTM.

Conclusion should conclude findings, not elaborate the study. It is written more like an abstract than conclusion. Please, shorten it I make conclusive remarks only!

Response:

The authors thank you for your comments for each part. We corrected and replied for each part here.

In introduction, some details which were not necessary, were deleted and added in results and discussion as following reviewer’s comments.

ASTM standard was cited in reference.

Conclusion part was rewritten.

Comments on the Quality of English Language

English should be improved. In some cases it is not understandable. Please use English proofreading to improve

Response:

The authors agreed for your comments. Thus, we used English proofreading service for this manuscript as attached certificate.

Reviewer 2 Report

Comments and Suggestions for Authors

1.       English sentences needs revision

2.       You cannot recycle clothes to virgin fibers. Consider revising, Ln 49

3.       Additional Features? Which other features? Include those features? Ln 53

4.       Sentence Ln 129 – 133 – long sentence and it makes no sense consider revising.

5.       “ creased”? ln 155 & 157 consider replacing with appropriate word.

6.       “from section 2.3 “ or from section 2.2” ln168

7.       Ln 381 – 387 is repeated in ln 393 – 399

8.       Figure 10  and figure 11 have been exchanged in the text.

9.       No color properties included in the results and yet the experiments section 2.3.6 reported that the color evaluation was done? 

Comments on the Quality of English Language

The English Needs Improvement

Author Response

Comments and Suggestions for Authors

Response:

The authors thanked you for reviewer’s comments and corrected for each point as shown in blue highlight in manuscript.

  1. English sentences needs revision

Response: The authors thank for reviewer’s comment. We used English proofreading service for this manuscript as attached certificate.

  1. You cannot recycle clothes to virgin fibers. Consider revising, Ln 49

Response: The authors agreed with reviewer’s comments. We revised this sentence.

  1. Additional Features? Which other features? Include those features? Ln 53

Response: The author added more details about additional features “for its intended use, such as breathability, stretchability, washability, or texture.

  1. Sentence Ln 129 – 133 – long sentence and it makes no sense consider revising.

Response: The authors revised that sentence to be short and clear. Fourier Transform Infrared (FTIR) analysis of the remaining PET was applied to characterize the functional groups of polymers within the cotton/PET textile waste blend before and after pretreatment.

  1. “ creased”? ln 155 & 157 consider replacing with appropriate word.

Response: The authors replaced stopped instead of creased for all manuscript.

  1. “from section 2.3 “ or from section 2.2” ln168

Response: The authors replaced from section 2.2 In 168.

  1. Ln 381 – 387 is repeated in ln 393 – 399

Response: The authors replaced stopped instead of creased for all manuscript.

  1. Figure 10  and figure 11 have been exchanged in the text.

Response: The authors corrected Figure 10 in the text.

  1. No color properties included in the results and yet the experiments section 2.3.6 reported that the color evaluation was done? 

Response: The authors deleted color properties in the experiments section 2.3.6.

Comments on the Quality of English Language

The English Needs Improvement

Response: The authors agreed for your comments. Thus, we used English proofreading service for this manuscript as attached certificate.

Reviewer 3 Report

Comments and Suggestions for Authors

The manuscript by Nanjaporn Roungpaisan et al is devoted to the production of PET fibers by the melt method and the production of fabrics from them. The title of the manuscript contains an abbreviation that will not be clear to most readers - "TC". The authors decipher this acronym in the abstract, but in my opinion it is better to use CP or PC instead of TC.
PET fibers have taken the largest place among the molded fibers. These fibers are in demand primarily due to their low cost, good mechanical and performance properties. Therefore, works devoted to this topic are relevant and in demand. In their introduction, the authors note the importance of PET fibers, methods of its recycling, and the difficulties associated with isolating the polymer from filament filaments. As an alternative to mechanical methods for isolating PET, it is proposed to use chemical methods using phosphoric acid. This acid dissolves cellulose and thus it is possible to separate the undissolved PET phase and cellulose fibers dissolved in the acid. Using thermal methods, the authors show that the use of acid is insufficient and it is proposed to use enzymes as an additional processing step. It has been shown that the use of enzymes makes it possible to achieve stable spinning of PET fibers when cellulose residues are present in the system; spinning is not stable and is accompanied by fiber breaks.

Abstract is written very chaotically and is difficult to understand. Perhaps this is due to its translation into English.
Keywords and their order require revision.

L.19, 20. "To make the remaining components contain, only pure polyester can be spun using the melt spinning method." I don't understand the meaning of this expression.
L.61. "Tetoron" is available here, I'll probably enter "Tetron"?
L. 123, 124. Why were these particular conditions chosen? As I know, the solubility of cellulose in phosphoric acid increases with decreasing temperature?!
L. 161-163. This information is duplicated and can be deleted.
L.180. The data given in this line does not agree with the data from table 1!
L. 204. "1.5" - it is not correct to round values. Authors must provide the full and common meaning of "1.54".
2.3.3. Thermogravimetric analysis (TGA). Why was this temperature range chosen? PET fibers are not used at high temperatures.
Figure 2. The methodological part indicates a range of up to 800 °C, and the figure shows a section up to 500 °C.
Figure 4 needs to be designed more carefully.
Figure 7. You need to add a scale bar to the figure.

The formatting of references (in the text) must comply with the rules of the journal.

The presented manuscript is executed at a good level, the main novelty of the work is concentrated in the production of PET fibers from recycled materials, the study of the structure and properties of these fibers. In my opinion, a correctly chosen mode of dissolving cellulose with acid can make it possible not to resort to the use of enzymes and, as a result, will simplify the processing of fabrics made from cellulose and PET.

Author Response

Comments and Suggestions for Authors

The manuscript by Nanjaporn Roungpaisan et al is devoted to the production of PET fibers by the melt method and the production of fabrics from them. The title of the manuscript contains an abbreviation that will not be clear to most readers - "TC". The authors decipher this acronym in the abstract, but in my opinion it is better to use CP or PC instead of TC.
PET fibers have taken the largest place among the molded fibers. These fibers are in demand primarily due to their low cost, good mechanical and performance properties. Therefore, works devoted to this topic are relevant and in demand. In their introduction, the authors note the importance of PET fibers, methods of its recycling, and the difficulties associated with isolating the polymer from filament filaments. As an alternative to mechanical methods for isolating PET, it is proposed to use chemical methods using phosphoric acid. This acid dissolves cellulose and thus it is possible to separate the undissolved PET phase and cellulose fibers dissolved in the acid. Using thermal methods, the authors show that the use of acid is insufficient and it is proposed to use enzymes as an additional processing step. It has been shown that the use of enzymes makes it possible to achieve stable spinning of PET fibers when cellulose residues are present in the system; spinning is not stable and is accompanied by fiber breaks.

Abstract is written very chaotically and is difficult to understand. Perhaps this is due to its translation into English.
Keywords and their order require revision.

Response: The authors thank you for pointing this out. We agree with this comment. Therefore, we corrected all typos as following of reviewer’s comments. The authors have already revision for abstract and keywords.

L.19, 20. "To make the remaining components contain, only pure polyester can be spun using the melt spinning method." I don't understand the meaning of this expression.

Response: The authors thank you for your comments. Therefore, we modified this expression.The remaining polyethylene terephthalate (PET) component was spun using the melt spinning method”.  

L.61. "Tetoron" is available here, I'll probably enter "Tetron"?

Response: The authors thank you for your comments. Both of Tetoron and Tetron Cotton were used for the combination of Polyester and Cotton fibers with combination of more polyester fiber compared to the cotton fiber. We used Tetron for TC as your comments.

  1. 123, 124. Why were these particular conditions chosen? As I know, the solubility of cellulose in phosphoric acid increases with decreasing temperature?

Response: The authors thank you for your comments. These particular conditions were chosen following method of reference 11.          

Shen, F.; Xiao, W.; Lin, L.; Yang, G.; Zhang, Y.; Deng, S. Enzymatic saccharification coupling with polyester recovery from cotton-based waste textiles by phosphoric acid pretreatment. Biores Technol 2013, 130, 248-255.

Also the solubility of cellulose in phosphoric acid increases with increasing temperature as cited by Zhang et al. (2009). As “high temperature accelerated the rate of cellulose dissolution in phosphoric acid,

the temperature of 50 °C was deemed an optimal condition for cellulose dissolution in phosphoric acid”.

Zhang, J.; Zhang, J.; Lin, L.; Chen, T.; Zhang, J.; Liu, S.; Li, Z.; Ouyang, P. Dissolution of microcrystalline cellulose in phosphoric acid-Molecular changes and kinetics. Molecules 2009, 14, 5027-5041. doi:10.3390/molecules14125027

  1. 161-163. This information is duplicated and can be deleted.

Response: The authors agreed for your comments. Thus, information in L161-163 with some duplicattion, was deleted.

L.180. The data given in this line does not agree with the data from table 1!

Response: The authors agreed for your comments. Thus, we corrected data in L184-185 agreed with the data from Table 1.

  1. 204. "1.5" - it is not correct to round values. Authors must provide the full and common meaning of "1.54".

Response: The authors agreed for your comments. Authors provided the full and common meaning of "1.54".

2.3.3. Thermogravimetric analysis (TGA). Why was this temperature range chosen? PET fibers are not used at high temperatures. The methodological part of TGA indicates a range of up to 800 °C, and the figure shows a section up to 500 °C.

Response: The authors thank you for your comments. 2.3.3. section is TGA of blended rPET fibers which was run into 800 °C in order to know degradation temperature and shown in Figure 9.

Figure 4 needs to be designed more carefully.

Response: The authors thank you for your comments. Figure 4 was redesigned more carefully.

Figure 7. You need to add a scale bar to the figure.

Response: The authors thank you for your comments. We added a scale bar to the Figure 7.

The formatting of references (in the text) must comply with the rules of the journal.

Response: The authors thank you for your comments. The authors corrected the references following the ref formatting of Polymers.

The presented manuscript is executed at a good level, the main novelty of the work is concentrated in the production of PET fibers from recycled materials, the study of the structure and properties of these fibers. In my opinion, a correctly chosen mode of dissolving cellulose with acid can make it possible not to resort to the use of enzymes and, as a result, will simplify the processing of fabrics made from cellulose and PET.

Response: The authors agreed for your comments. However, in case of recycle of PET from TC fabrics, the cellulose must not remain in the PET. If not, during compression process, the remained cotton will degrade to dark in the component which affects to melt spin of PET. Thus, enzyme treatment is needed to apply after phosphoric treatment.

Round 2

Reviewer 3 Report

Comments and Suggestions for Authors L. 395, 399, 411. I recommend that the authors change (paraphrase) the beginning of each paragraph (sentence). L. 467. "usable" - I recommend removing it. It is necessary to correct the conclusions - remove the highlight (bold), delete - "5. Patents".